# Effects of a Novel Adsorbent on Membrane Fouling by Natural Organic Matter in Drinking Water Treatment

**DOI:** 10.3390/membranes9110151

**Published:** 2019-11-12

**Authors:** Lelum D. Manamperuma, Eilen A. Vik, Mark Benjamin, Zhenxiao Cai, Jostein Skjefstad

**Affiliations:** 1Aquateam COWI, P. O. Box 6412, 0579 Oslo, Norway; 2MicroHAOPs Inc., 4000 Mason Road, Seattle, WA 98195, USA; mark.benjamin@microhaopsinc.com (M.B.); nathan.Cai@microhaopsinc.com (Z.C.); 3Ullensaker Municipality, Furusethgt. 12, 2050 Jessheim, Norway; Jostein.Skjefstad@ullensaker.kommune.no

**Keywords:** adsorption, HAOPs, NOM, MF anti-fouling, membrane filtration

## Abstract

Irreversible fouling of water filtration membranes reduces filter longevity and results in higher costs associated with membrane maintenance and premature replacement. The search for effective pretreatment methods to remove foulants that tend to irreversibly foul membranes is ongoing. In this study, a novel adsorbent (Heated Aluminum Oxide Particles (HAOPs)) was deployed in a fully automated pilot system to remove natural organic matter (NOM) from the surface water source used at the UniVann water treatment plant (WTP) in Ullensaker County, Norway. The pilot plant treatment process consists of passing the water through a thin layer of HAOPs that has been deposited on a mesh support. The HAOPs layer acts as an active packed bed which removes NOM from the water. Fluxes around 120 L/m^2^/h (LMH) at transmembrane pressure (TMP) below 10.7 psi (0.7 bar) were achieved over production cycles excessing 12 h. Treatment achieved always >85% colour removal and effluent colour <5 mg Pt/L (the target of treatment), and always <0.01 NTU turbidity and non-detectable suspended solids in the permeate. The HAOPs mixture after saturated with NOM is easy to remove by disruption of the HAOPs by rinsing the mesh surface, and the sludge is easily dewatered to higher of dry solids content.

## 1. Introduction

Clean drinking water is one of the basic needs for human beings especially under extreme conditions. Regular sources of drinking water are often polluted by harmful and hazardous components which can cause considerable losses of life, especially in the aftermath of natural disasters. Other contaminants have less acute effects but are of concern because of long-term health impacts or economic or aesthetic considerations.

In recent years, the treatment requirements for higher quality drinking water are getting stricter and is expanding to include a wide variety of residual chemicals, micropollutants and other contaminants. Drinking water sources are becoming limited and more polluted sources (rivers and seawater/brackish water) are included. Membrane filtration (MF) and reverse osmosis (RO) processes are increasingly becoming used. Natural Organic Matter (NOM) is present in surface water and seawater/brackish water and causes a variety of problems including aesthetic (colour, turbidity, taste and odour), health (organic halogens or by acting as a vehicle for micropollutants) and operational by reducing the efficiency or capacity of water treatment processes [1]. When water is disinfected with chlorine, NOM in the water is also producing by-products such as trihalomethanes (THMs), haloacetic acids (HAAs), and cyanogen halides (CNXs) [2]. Effective removal of NOM is therefore desirable, and a wide range of technologies are applied Coagulation followed by separation (sedimentation, flotation and/or filtration) is the most commonly used process, but membrane processes (MF), and ozonation followed by activated carbon (biological activated carbon; BAC) is also applied. 

The most frequently applied process for removal of NOM includes using coagulation. The efficiency of this process depends on the type and the dose of coagulants, the solution pH, and the NOM characteristics. Typically, higher coagulant doses are required for NOM removal than for turbidity removal, and there is normally a stoichiometry in the required coagulant dose with NOM concentration. In recent years, low-pressure membrane systems (e.g., microfiltration (MF)) have proven to be a promising technology [3,4] for replacing conventional technologies such as coagulation, flocculation [4], contact filtration [5] and ozonation [6]. 

MF offers many advantages compared to conventional methods, including high quality effluent, reliability in operation, simple operation conditions, high output production with lower energy and chemicals consumption, and a small footprint. However, low-pressure MF systems also have several disadvantages that limits their use. One critical drawback of membrane technology for surface water treatment is membrane fouling. Fouling reduces membrane permeability and productivity over time, which increases energy consumption, reduces the amount of water processed, increases the frequency of chemical cleaning, and reduces membrane lifetime [4]. Furthermore, fouling results in a buildup of suspended and colloidal matter, organic matter, inorganic material, and biota on membrane pores and surfaces. This process can lead to irreversible loss of permeate flux through the membrane [7,8] 

Researchers have proposed various approaches for minimizing membrane fouling, including physical, chemical and hydrodynamic methods such as mechanical cleaning, modification of the membrane surface to minimize interactions between the membrane and the deposits, improve module design and reduce membrane loading rate (LMH) and NOM loading (TMP) in order to reduce NOM deposition on the membrane pores and surface [4]. However, many of these approaches consume substantial energy, chemicals and reduces capacity of the membranes and has limited MF to be widely implemented in large-scale water treatment plants. Kim, et al. [9] studied an alternative method to remove NOM and thereby control membrane fouling, by using a novel adsorbent: heated aluminum oxide particles (HAOPs). They proposed a pre-treatment ultrafiltration membrane step including removing NOM by HAOPs pre-deposited onto the surface of a membrane installed upstream the MF. 

Early research on HAOPs documented reductions in membrane fouling and THM formation potential by pre-deposited HAOPs [10]. Subsequent work showed that, if raw water is filtered through a thin, pre-deposited layer of HAOPs upstream of a membrane, substantial amounts of NOM can be removed from the water, thereby reducing the fouling rate of downstream membrane units [11,12,13,14,15,16].

The current study is based on joint research initiated between MicroHAOPs and Aquateam COWI with the objective of optimizing the upstream pre-deposition HAOPs treatment step into a fully automated pilot system and document the benefits and potential for industrializing the process. Collaboration was established with UniVann in 2014. Until the end of 2018 the waterwork was operated by Ullensaker and Nannestad county and delivered water to ~35,000 pe. This work describes a possible solution to optimize the existing direct filtration process at UniVann drinking water treatment plant. The main problem in the direct filtration process is the optimal coagulant dose, which is mainly dependent on parameters of raw water quality that includes the content of particles and colour produced by NOM. Over dose of coagulants may leads for health risks, increase the amount of sludge and sludge treatment costs. Furthermore, coagulation is used as a pre-treatment for the membrane filtration to reduce membrane fouling. Both the cases require a very good dosing control system to keep the required effluent water quality regardless the sudden changes of raw water quality. HAOPs pre-treatment technology addresses above questions with automation and operation of the technology, initial experiences gained and compares with the existing traditional direct filtration plant and experiences from MF for NOM removal. 

## 2. Materials and Methods

MicroHAOPs Inc. developed a fully automated user-friendly pilot system with 2 Mesh filtration modules. Each module contains 69 nylon tubes. Prior to filtration, the HAOPs are pre-deposited onto the inside surface of these tubes. The pilot system used online colour (Chemtrac UVM5000, Norcross, GA, USA), turbidity (MicroTol 40060, Watts, North Andover, MA, USA) and TMP as indicators of water quality and process performance. Operational parameters such as HAOPs dosage, backwash time and number of backwashes and filtering cycle length, can be chosen by the user. The pilot unit can be operated with either module alone, or with both modules active; in the latter case, the modules can be arranged in parallel or in series. Data is logged at 3 min. intervals, and full access and control of the pilot unit is enabled from anywhere. 

All chemicals were of reagent quality. HAOPs containing about 28% Al by mass were synthesized by heating precipitated Al(OH)_3_, following a previously described procedure [12]. Figure 1 shows a schematic diagram of the pilot system. The two HAOPs mesh filtration modules, are 120-mm-diameter Both the HAOPs and the raw water is entering from the bottom of the modules and back wash water enters from the middle (perpendicular to the tubes) and flush out from top and bottom of the modules. Each pump has its own solenoid valve and non-return valve. The control system consisted of pumps, valves, UV, turbidity, flow and pressure detectors. The pressures upstream and downstream of the tubes were continuously monitored, the effluent UV-absorbance and TU were measured on-line, and the system Programmable Logic Controller (PLC) was connected to a computer through an interface. Control system can be easily connected to the Supervisory Control and Data Acquisition (SCADA) system as well as can be operated by remote access system.

Beginning of a filtering cycle starts with dosing and pre-coating of HAOPs inside the mesh tubes. Pumps, valves and other units will be start/stop according to the HAOPs dosage. Raw water filtering from inside to outside of the mesh tubes, where the HAOPs layer works as the adsorbent. The filtering cycle is active until the system reaches one of the set points. The main set points are TMP, effluent colour and turbidity. The backwashing cycles will be activated, if the system reaches one of the set points, until that, the filtering cycle is active. The backwash cycle is followed by the flush cycle, where an air compressor supplies air into the bottom of module, through a porous diffuser, at a rate of approx. 20–25 psi. Same operation sequence is used independent of operation of the modules in series or parallel mode of operation. The only difference for series operation is that modules are shifting from upstream to downstream after each backwash cycle. 

The pilot plant is equipped with sensors, connected to PLC, personal computer (PC) and a data logging system, recording the following parameters: HAOPs flow, raw water flow, TMP, inlet and effluent water quality (colour and turbidity), permeate flux and air flow. All input and output signals/data are displayed online to monitor system functioning and record in a PC memory with a selectable time stamp for future processing. In summary, the new pilot unit, described above, has several innovative features and distinct advantages for effective water treatment, and is simple to operate. It is fully automated and operate unmanned for long time-periods using remotely monitored information through a novel wireless communication system. Continuous operation is implemented with limited reject streams (that would need further care), thus permitting high clean water recovery. Although the system has not been optimized yet, the energy consumption appears to be relatively low, compared with conventional water treatment processes, thus rendering the unit a promising cost/effective and environmentally friendly water-treatment technology.

The pilot plant was installed at the UniVann water treatment plant in Ullensaker county, Norway. The full-scale treatment process at UniVann is a direct filtration plant (chemical coagulation with an iron-based coagulant (PIX-318) followed by dual media filters) followed by alkaline media filters for corrosion control (pH, alkalinity and Ca adjustment) and disinfection by UV and chlorination. UniVann’s capacity was (until the end of 2018) around 14,000 m³/day, the colour of the raw water varies from 40–60 mg Pt/L, and the treated water target was to meet the requirement of colour <5 mg Pt/L. 

Ultraviolet absorbance at 254 nm (UV254) was used to quantify the concentration of organics in the water. UV254 was measured using a spectrophotometer (Hach-Lange DR 3900, Loveland, CO, USA) with a 5-cm quartz cell. UV254 is converted to mg Pt/L by a developed correlation curve between UV254 and colour based on absorbance at 410 nm wavelength. The Particle Size Distribution of HAOPs was measured by Malvern Mastersizer 3000 (Malvern, UK), a laser diffraction monitoring method. 

## 3. Results and Discussion 

The Particle size distribution (PSD) cumulative curves of HAOPs samples made in 14 different batches during the test period are shown in Figure 2. All PSD cumulative curves have similar characteristics, with in average 15% of the HAOPs particles < 10 µm. The PSD cumulative curves confirms, that the synthesis of HAOPs in different laboratories does not affect the quality of HAOPs particles. Based on the HAOPs size, the pre-deposition treatment technology was constructed using a 10 µm nylon mesh tube, and the HAOPs was pre-deposited in a recycling mode to limit loss of HAOPs. 

Figure 3 shows water quality data from UniVann water treatment plant in 2018. Incoming (raw) water colour is varying between 30 and 55 mg Pt/L, and treated water is for most of the time meeting the target of <5 mg Pt/L, but was varying between 5 and 10 mg Pt/L, with one peak up to 20 mg Pt/L. The water temperature varies from 4 to 17 °C over the year. The variation of raw water quality at UniVann is influenced by temperature, precipitation and snow melting conditions. Increased frequency and severity of extreme events such as floods and storms are increasing the variations. In periods with high precipitation, alteration in hydrological flow patterns might occur. Enduring rain causes saturated soils with increasing flow through upper organic soil horizons. This causes changes in the fluxes and chemical composition of NOM compared to low flow conditions. Changes in the quantity and quality of dissolved NOM will subsequently influence physical and chemical parameters in UniVann’s water source (Lake Bjertnessjøen). The colour of raw water to UniVann increased from 20 in 1994 to 55 mg Pt/L in 2018.

Furthermore, Norwegian waterworks and health authorities are experiencing a significant increase in water colour of lakes and rivers used for drinking water. Some of the largest Norwegian waterworks have experienced more than a doubling of raw water colour during last decade. Increasing water colour is of hygienic and economic importance since it interferes with common disinfection procedures and necessitates more comprehensive water treatment and automation to comply with drinking water standards. Direct filtration and alkaline media filters are the most common Norwegian water treatment process, and for the New-Ullensaker water treatment plant, which started up in 2019, this process were also selected.

Typical challenges experienced for NOM removal at the direct filtration process applied on UniVann was:Challenges to atomize the chemical treatment process, since the NOM characteristics changes over time, the temperature varies and the very low winter temperature requires careful chemical treatment, and during rainfall the NOM concentration varies quickly. This requires a very good dose/control systems for coagulantsSubstantial loss of water since the media-filters low turbidity and provide an acceptable hygienic barrier quality for treated water (turbidity <0.1 NTU).Controlling metal loss from the coagulant to treated drinking water (<0.1 mg Al/L from aluminium based coagulants such as Polyaluminium Chloride (PAC)) and to local rivers/lakes/creeks. Experiences gained at UniVann was that high reject flows from dewatering of chemical sludge gave water with high residual Al-concentrations needing control to avoid killing fish. Therefore PIX was chosen, but loss of Fe into the drinking water distribution system would form particulates and deteriorate quality (turbidity) of water in the distribution system.Simplifying water treatment plant sludge handling since direct filtration forms large volumes of reject from backwashing of water, and the sludge from this is difficult to dewater. Centrifuges was used and polymers are needed to produce sludge with a dry solids content ~13% was achieved.

Figure 4 shows results from initial testing of the pilot plant with the two HAOPs treatment modules operated in parallel mode at a filtration flux of 120 LMH with and without pre-deposited HAOPs. In this experiment 15 g Al/m^2^ of HAOPs was pre-deposited on one module, while the other module was not added any HAOPs. Both modules have a 10 µm nylon microsieve. The filtering period was 3 h for both modules. When the filtering period was finished, both the modules were backwashed and a new filtering cycle initiated. For each cycle, the module with pre-deposited HAOPs exhibited a lower rate of TMP rise (Diff Press 2) than the module without HAOPs (Diff Press 1). 

The TMP in the module without HAOPs increased during the experiment, whereas that in the module containing HAOPs did not. Thus, it appears that the permeability of the tubes in the former system was significantly influenced by irreversible fouling and could not be recovered by backwash (rinsing). By contrast, in the system with pre-deposited HAOPs, the fouling that occurred during a cycle was reversed by the same physical rinsing steps. No effort was made to optimize the operation, and it is likely that the system could have been operated with longer intervals between HAOPs dosing without performance deterioration.

The nylon mesh with pre-deposited HAOPs had given rise to the remarkable improvement in lower TMP buildup of meshes, which might be attributed to the reduced the chemical cleaning frequency. Furthermore, the HAOPs layer can be treated as a packed bed of adsorbent, and the meshes does not become fouled until the NOM breaks through the HAOPs layer and get in contact with the mesh itself. Once such breakthrough occurs, fouling progresses rapidly. The NOM-foulant that is removed by the HAOPs never contacts the meshes, so when the HAOPs layer is backwashed and removed from the system, very little NOM remains on the mesh, suggesting that backwashing should be a very effective approach for cleaning the mesh, so that the hydraulically irreversible fouling may reduce by proper backwash cycle. 

Figure 5 illustrates the NOM removal with and without pre-deposited HAOPs on the membrane surface. At any given specific volume treated (the volume of permeate produced per unit area of membrane) a significant amount of NOM was removed when HAOPs was pre-deposited on the membrane (effluent colour from all the runs was <10 mg Pt/L), even though some NOM passed through the HAOPs layer and appeared in the permeate at the beginning of each run. The result was consistent with previous findings of Kim and Cai [12,14], that HAOPs efficiently removed colour, suggesting that they selectively bind or trap more (NOM) hydrophobic molecules [17]. 

In principle, as long as fouling NOM molecules are removed by the HAOPs and do not come in contact with the supporting mesh, the mesh is protected from fouling. The experimental data suggests that fouling and poor colour removal occurs only after a significant amount of NOM breaks through the HAOPs layer and reaches the supporting tube. 

In the next experiments the modules were operated in series mode with HAOPs dosages of 30g Al/m² to each module, a filtration flux rate of 100 LMH and a pre-set operating time between backwashing of 12 h. The idea behind operation of the HAOPs pre-deposited nylon mesh units in series is to optimize the chemical usage, just like you would operate granular activated carbon (GAC) filters to optimize the GAC usage. The first in the series can operate at higher effluent colour since the second will reduce the colour to <5 mg Pt/L. The limiting factor, could, however, be that too high NOM concentrations is reaching the nylon surface and will increase fouling of the nylon mesh so that chemical treatment (NaOH) is needed more often.

The series operational mode functions as follows. First cycle, HAOPs were pre-deposited in both modules, with module 1 (M1) upstream and module 2 (M2) downstream. After the filtering period, only M1 was backwashed and a new layer of HAOPs was pre-deposited in M1, while feed was injected directly into M2. After deposition of fresh HAOPs in M1, effluent from M2 was directed to M2; i.e., M2 functioned as the upstream module and M1 as the downstream module. After the next filtering period, only M2 was backwashed and re-deposited with HAOPs. M2 was then receiving the effluent from M1, as in the initial cycle. The whole process was then repeated multiple times, while the direction of the water flow was alternating between having M1 or M2 as the first and second unit in the series. 

Figure 6 illustrates the effluent water quality during this experiment. In the first cycle (M1 to M2), effluent colour at the beginning of the filtering cycle was 2.5 mg Pt/L, and after 12 h it reached 15 mg Pt/L, while the turbidity was stable at 0.01 NTU. In the second cycle (M2 to M1), effluent colour began at 11 mg Pt/L and after 12 h it reached 24 mg Pt/L while the turbidity was stable at 0.01 NTU. The continuous increase of colour in the effluent, indicates that too much NOM has been removed by the HAOPs and that the nylon mesh was not sufficiently cleaned between the treatment cycles. This optimization of operation is yet not completed. 

In general, the permeate water quality was quite consistent regardless the raw water quality. In average the treatment efficiency showed >83% colour removal. Suspended solids could not be detected in the permeate and the permeate turbidity was <0.01 NTU throughout the whole experiment. The permeate was also tested for residual aluminum and all samples showed less than 0.035 mg/L. The final permeate water quality was well within the requirements of the Norwegian drinking water standards. Finding optimum LMH, TMP and HAOPs loading per area to maintain operational stability, optimum HAOPs consumption and cost/efficient operational window are present R&D focus. Modifications are presently made to the backwashing system, and results are promising.

## 4. Conclusions 

The multi-cycle filtration tests showed that a pre-deposited HAOPs layer can reduce fouling while removing substantial NOM from a drinking water source, suggesting that this technology has potential benefits for long-term operation in practical applications. 

Adsorption of the NOM onto HAOPs adsorbents can remove colour from the water, and the NOM-HAOPs layer can be easily separated from the mesh surface after the filtration step, keeping the mesh surface physically clean.

Future work will include investigating operating cycles, a better understanding of the effect of the cleaning cycles on the system performance, improving the reuse/ recovery of HAOPs and optimizing modes of operation to minimize fouling. 

## Figures and Tables

**Figure 1 membranes-09-00151-f001:**
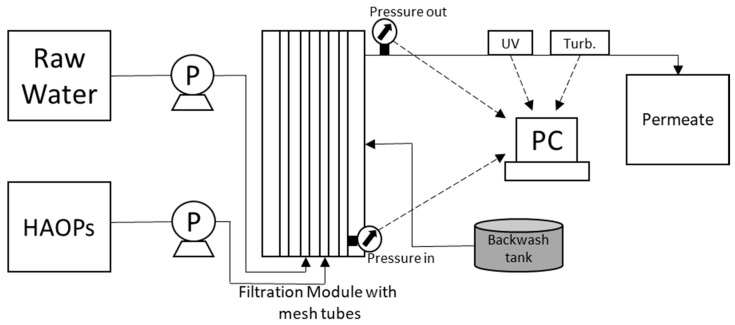
Schematic diagram of the pilot system.

**Figure 2 membranes-09-00151-f002:**
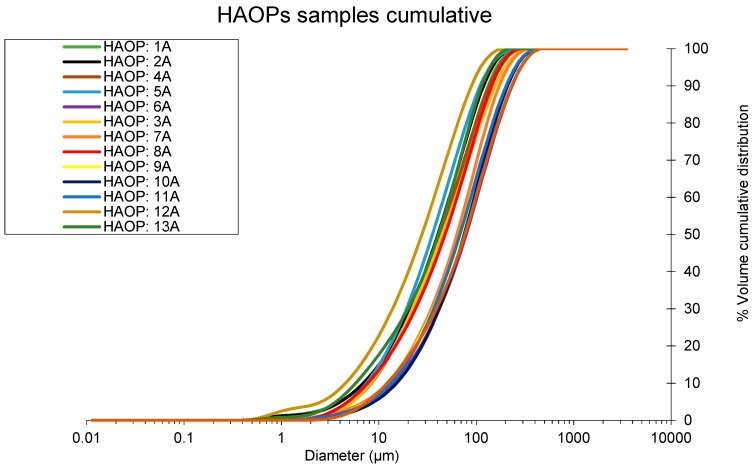
Particle size distribution cumulative curves of 14 different batches of HAOPs samples prepared during pilot testing.

**Figure 3 membranes-09-00151-f003:**
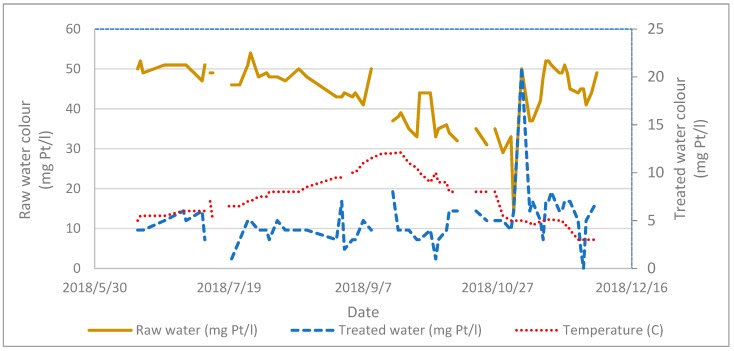
The existing waterwork at UniVann’s raw- and treated water quality (colour) and temperature from May through December 2018.

**Figure 4 membranes-09-00151-f004:**
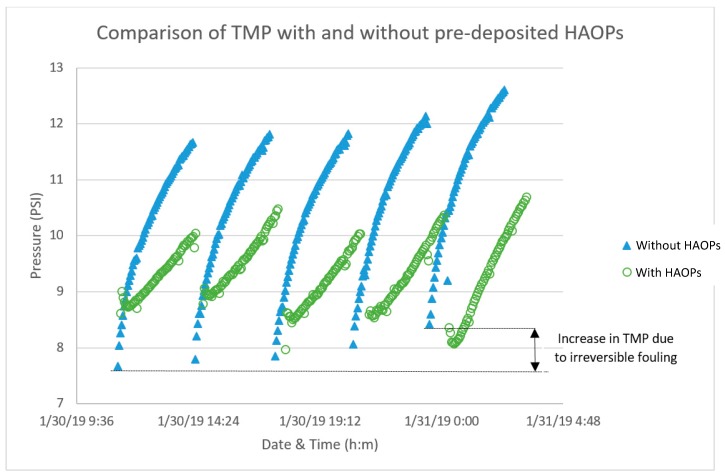
Comparison of initial and increased start TMP over several filtering cycles at 120 LMH without 15 g Al/m² (Blue ∆, module 1) and with (Green O, module 2) pre-deposited HAOPs in the tubes; constant filtering period 3 h between back washing.

**Figure 5 membranes-09-00151-f005:**
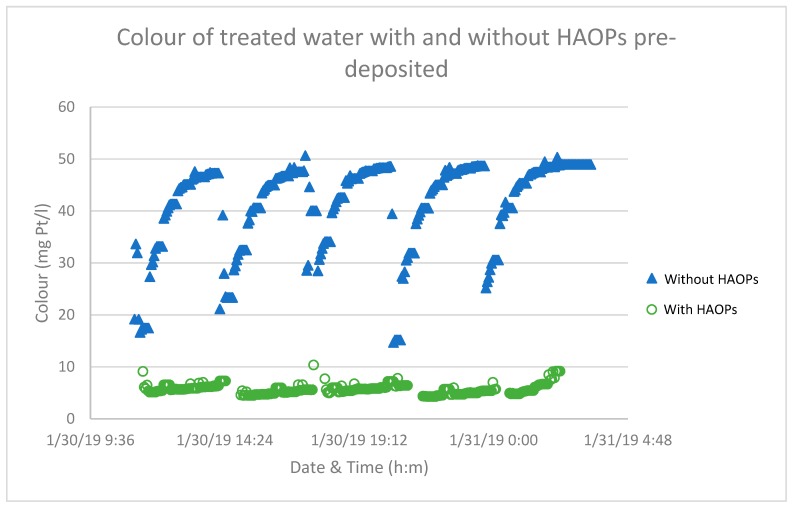
Comparison of colour of treated water in each filtering cycle without (∆) and with (O) pre-deposited HAOPs.

**Figure 6 membranes-09-00151-f006:**
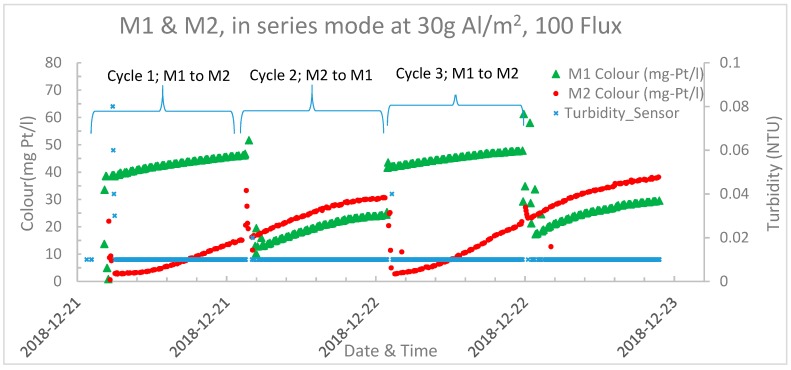
Effluent turbidity and colour at HAOPs dosage of 30 g Al/m² and Flux 100 LMH; inlet raw water colour 60 mg Pt/L.

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
