# Peer review of "Effects of a Novel Adsorbent on Membrane Fouling by Natural Organic Matter in Drinking Water Treatment"

_membranes, 2019, doi:10.3390/membranes9110151_

Round 1

Reviewer 1 Report

This study demonstrates the application of heated aluminum oxide to lower fouling propensity of membrane filtration and enhance removal efficiency of natural organic matter. The results of reduced fouling potential and enhanced color removal of raw water are promising and interesting. However, the manuscript is likely to be a technical report on a pilot scale study. More experimental details and relevant mechanisms are recommended to be incorporated to provide a more useful scientific report.

Abstract: “Fluxes around 120 LMH at TMP below 10.7 PSI (0.7 bar)…”: please to more specific by providing more details about types of membrane used and information about feed (waste) stream. Please provide details for the removal mechanisms of natural organic matters by the heated aluminum oxide particles. Why AOPs are heated before being applied on the mesh? Does heating increase the NOM removal efficiency? What is an outstanding features of aluminum to remove colour/NOM compare to other oxide-based metals Water quality monitoring: The study evaluated the efficiency of HAOPs by measuring the colour of the treated water. Question arises whether the colour is the only and enough parameter determining the NOM removal efficiency and/or removal of colour can directly to be translated to the removal rates of NOM? Economics of HAOPs loading? This study seems to target practical application of HAOPs. How’s the economic feasibility in terms of the amount of HAOPs used for every filtration cycles? Also, are HAOPs recyclable? Figure 4-5: Revise the labels on the right side of the Figure “xxx_1” and “xxx_2” to “xxx with HAOPs” and “xxx without HAOPs”

Author Response

Response to Reviewer 1 Comments

Thank you for your letter and the opportunity to revise our paper on ‘Effects of a novel adsorbent on membrane fouling by natural organic matter in drinking water treatment.’ The suggestions offered by the reviewers have been immensely helpful, and we also appreciate your insightful comments on revising the abstract and other aspects of the paper.

I have included the reviewer comments immediately after this letter and responded to them individually, indicating exactly how we addressed each concern or problem and describing the changes we have made. The revisions have been approved by all four authors and I have again been chosen as the corresponding author. The changes are marked in text, and the revised manuscript is submitted in the Journal web page.

Abstract: “Fluxes around 120 LMH at TMP below 10.7 PSI (0.7 bar)…”: please to more specific by providing more details about types of membrane used and information about feed (waste) stream.

According to your comments,, more information about the feed water quality (figure 3 and text) was included in the text and type of mesh used (Based on the HAOPs size, the pre-deposition treatment technology was constructed using a 10 µm nylon mesh tube, and the HAOPs was pre-deposited in a recycling mode to limit loss of HAOPs.) was included in the text.  

Please provide details for the removal mechanisms of natural organic matters by the heated aluminum oxide particles. Why AOPs are heated before being applied on the mesh? Does heating increase the NOM removal efficiency? What is an outstanding features of aluminum to remove colour/NOM compare to other oxide-based metals Water quality monitoring:

More details are included in the "introduction" section. Early research of HAOPs were done by MicroHAOPs team.

Kim  (Kim et al. 2007) studied an alternative method to remove NOM and thereby control membrane fouling, by using a novel adsorbent: heated aluminum oxide particles (HAOPs). They proposed a pre-tretament ultrafiltration membrane step including removing NOM by HAOPs pre-deposited onto the surface of a a membrane installed upsteam the MF.

            Early research on HAOPs documented reductions in membrane fouling and THM formation potential by pre-deposited HAOPs (Wang and Benjamin 2016). Subsequent work showed that, if raw water is filtered through a thin, pre-deposited layer of HAOPs upstream of a membrane, substantial amounts of NOM can be removed from the water, thereby reducing the fouling rate of downstream membrane units (Malczewska and Benjamin 2016; Cai, Kim, and Benjamin 2008; Cai and Benjamin 2011; Kim, Cai, and Benjamin 2008; Kim, Cai, and Benjamin 2010; Kim, Deng, and Benjamin 2008).

The study evaluated the efficiency of HAOPs by measuring the colour of the treated water. Question arises whether the colour is the only and enough parameter determining the NOM removal efficiency and/or removal of colour can directly to be translated to the removal rates of NOM? Economics of HAOPs loading?

For this study used a co-relation between colour-254UV absorbent to measure efficiency of HAOPs. Next stage of the study is to combine more parameters to evaluate HAOPs as well as to further optimization of HAOPs dose, sludge quality and sludge treatment methods, re-use/recovery of HAOPs and re-use of NOM, HAOPs-NOM bond/adsorbing strength, energy consumption for synthesis of HAOPs. These studies are under the laboratory experiments.   

This study seems to target practical application of HAOPs. How’s the economic feasibility in terms of the amount of HAOPs used for every filtration cycles? Also, are HAOPs recyclable? Figure 4-5: Revise the labels on the right side of the Figure “xxx_1” and “xxx_2” to “xxx with HAOPs” and “xxx without HAOPs” 

Corrected according to comments

Reviewer 2 Report

The paper deals with the use of Heated Aluminum Oxide Particles [HAOPs] in a fully automated pilot system to remove NOM from the surface water source used in a water treatment plant.

The manuscript well describes the pilot system and the results achieved in NOM removal are interesting, therefore the paper deserves publication.

I have only few questions to submit to the authors:

1) It is mentioned the energy consumption in this system appears relatively low with respect to others, this aspect could be further developped in order to evaluate the real economic impact of the plant (also considering that a sludge is produced at the end of the treatment cycle).

2) The authors state that the NOM adsorption on mesh not covered by HAOPs produces irreversible fouling. Do they have direct proof of this statement?

3) In the abstract is reported that "the sludge is easily dewatered to >30% dry solids content": I did not find any detail of this aspect in the text, maybe useful to include it in the manuscript body.

An important aspect to modify in the text concerns the particle size distribution. Actually the curves reported are not distributions but cumulative curves, the derivative curve should be inserted to show the particle size distribution.

Some minor points:

Do not cite the page interval of the references in the text.

Full stops to add: line 42, 107.

line 112: define PLC

line 131: "can be operated" instead of "operate"

line 144: an s appears in the sentence with no apparent meaning

line 161, 202: errors related to reference to figures

Fig.3: modify the x-scale values. In the same figure, where is the T scale?

line 185: "were" instead of "was"

line 186: "change" instead of "changes"

line 190 and following: Check the meaning of the second and forth point of the list.

line 193: define PAX and close bracket.

line 222: check the meaning

line 229: minimal?

Author Response

Response to Reviewer 2 Comments

Thank you for giving me the opportunity to submit a revised draft of my manuscript titled "Effects of a novel adsorbent on membrane fouling by natural organic matter in drinking water treatment" to Membrane journal. I here appreciate the time and effort that you and the reviewers have dedicated to providing your valuable feedback on our manuscript. We are grateful to the reviewers for their insightful comments on our paper. We have been able to incorporate changes to reflect most of the suggestions provided by the reviewers. We have highlighted the changes within the manuscript. Here is a point-by-point response to the reviewers’ comments and concerns

I have only few questions to submit to the authors:

1) It is mentioned the energy consumption in this system appears relatively low with respect to others, this aspect could be further developped in order to evaluate the real economic impact of the plant (also considering that a sludge is produced at the end of the treatment cycle).

System is on the pilot stage; next stage is to optimize HAOPs dosage, cycle time and target more study's on the energy consumption and economic impact. After each cycle, used-HAOPs removed from the mesh as sludge, Sludge is a combination of HAOPs and NOM mixture. Possibility of re-use of HAOPs and NOM will be one of the targets in the next stage of the pilot system.  

2) The authors state that the NOM adsorption on mesh not covered by HAOPs produces irreversible fouling. Do they have direct proof of this statement?

This statement was formed by the figure 4 further proof is in the stage of laboratory stage. According to the figure 4, tests were conducted with and without HAOPs. TMP of the mesh with-HAOPs (in each cycle) begins at around 8.5 PSI and mesh without-HAOPs it begins (first cycle) 7.5 PSI and the last cycle 8.2 PSI. Both the modules were back-washed with the same conditions, but the TMP of the mesh without-HAOPs increases in each cycle, this is the reason for the above statement.     

3) In the abstract is reported that "the sludge is easily dewatered to >30% dry solids content": I did not find any detail of this aspect in the text, maybe useful to include it in the manuscript body.

Changed in the text of abstract.

An important aspect to modify in the text concerns the particle size distribution. Actually the curves reported are not distributions but cumulative curves, the derivative curve should be inserted to show the particle size distribution.

PSD curves has been change to cumulative curves in the text.

Some minor points:

Do not cite the page interval of the references in the text.

Full stops to add: line 42, 107.

line 112: define PLC

line 131: "can be operated" instead of "operate"

line 144: an s appears in the sentence with no apparent meaning

line 161, 202: errors related to reference to figures

Fig.3: modify the x-scale values. In the same figure, where is the T scale?

line 185: "were" instead of "was"

line 186: "change" instead of "changes"

line 190 and following: Check the meaning of the second and forth point of the list.

line 193: define PAX and close bracket.

line 222: check the meaning

line 229: minimal?

The above mention "minor points" has been corrected according to the reviewer's comments.   
